

# Diagnosing ice sheet grounding line stability from landform morphology

Lauren M. Simkins[1*], Sarah L. Greenwood[2*], John B. Anderson[1]

[1] Department of Earth, Environmental, and Planetary Sciences, Rice University, Houston, TX 77005, USA
[2] Department of Geological Sciences, Stockholm University, 10691 Stockholm, Sweden
*Equal contributions

*Correspondence to*: Lauren M. Simkins (lsimkins@rice.edu)

**Abstract.** Ice sheet grounding lines not only define where an ice sheet flux meets and interacts with the ocean, but also represent sedimentary environments, where an upstream sediment flux reaches the ice sheet margin. Landforms that form at the grounding lines hold the potential to reveal the nature of the processes that govern this dynamic and potentially vulnerable environment. Here we analyse a large dataset (n=6,275) of grounding line landforms mapped on the western Ross Sea continental shelf from high-resolution geophysical data. Their morphometric properties divide the population into two distinct morphotypes: recessional moraines (consistently narrow, closely spaced, low amplitude, symmetric, and straight), and grounding zone wedges (broad, widely spaced, higher amplitude, asymmetric, sinuous, and highly variable). Landforms transition abruptly between morphotypes, both spatially along a continuous grounding line position and temporally within a retreat sequence. We find minimal effect of water depth or topography on the production of one landform or the other, and find no conclusive evidence for morphology being determined by the presence or absence of an ice shelf. Instead, we find that both sediment supply to the grounding line and the time for which a grounding line is occupied are important in determining the resultant landform morphology. The development of grounding zone wedge asymmetry through sediment progradation representing longevity of a grounding line position ('stable'), while the development of sinuosity due in part to basal meltwater flushing of sediment through grounding line embayments is linked with large magnitude retreat events ('unstable'). We find that while longer duration grounding line positions form grounding zone wedges and are destabilised in the form of larger magnitude retreat, short-lived grounding line positions manifest as recessional moraines back-step with small magnitude retreat events. These two resulting retreat styles appear to reflect differences in sensitivity to processes that control grounding line retreat both in space and time.

## 1 Introduction

Marine-based ice sheet stability is strongly influenced by perturbations near the grounding line, the downstream most location grounded ice is in contact with the underlying bed (e.g., Schoof 2011; Robel et al., 2014). The grounding line position is fundamentally determined by ice thickness relative to water depth, where ice is sufficiently thick to overcome buoyancy (Fig. 1A), and where ice thickness in turn is determined by mass balance at the grounding line. A broad suite of



processes and conditions that locally dictate both buoyancy and mass balance make it difficult to reliably distinguish and define grounding line positions as 'stable' versus 'unstable'. Yet predicting how ice sheet sectors will respond to their grounding lines being dislodged by enhanced melt or rising sea level under future warming scenarios or, conversely, how grounding lines will respond to changes in interior ice flow behaviour, is an urgent endeavor.

The flux of ice to ice sheet grounding lines is highly spatially variable, determined by the overall flow structure of the ice sheet (Bamber et al., 2000; Rignot et al., 2011), its basal thermal regime (Kleman and Glasser, 2007), basal slipperiness due to the distribution and style of meltwater drainage (Stearns et al., 2008), cyclic responses of subglacial till rheology to tides (Doake et al., 2002; Anandakrishnan et al., 2003; Gudmundsson, 2007), and the effects of ice shelf buttressing (Rignot et al., 2008; Hulbe et al., 2008). Mass loss occurs by calving and by sub-marine melting of the ice front and ice shelf, the balance

between which can vary enormously, with orders of magnitude variability in melt rates (Depoorter et al., 2013; Rignot et al., 2013). Ocean-driven basal melting of ice shelves is thought to be concentrated near grounding lines (e.g., Jenkins and Doake, 1991; Rignot and Jacobs, 2002), and channelised subglacial freshwater emanating at grounding lines can lead to locally enhanced ice shelf melting (Le Brocq et al., 2013; Marsh et al., 2016). While the magnitude of these processes and changes therein may predispose an ice sheet grounding line to advance or retreat, the *position* of the grounding line - and,

one might expect, the duration with which it is held - is dictated by the buoyancy of ice. Since water depth is a primary control, grounding line position may be sensitive to sea level change (Thomas and Bentley, 1978; Schoof, 2007; Katz and Worster, 2010), the modulating effects of glacial-isostatic adjustment (Gomez et al., 2010), as well as bed topography - either in the form of antecedent topography (e.g., Jenkins et al., 2010; Matsuoka et al., 2015; Halberstadt et al., 2016) or the sedimentary construction of relief at the grounding line itself (Anderson, 1999; Alley et al., 2007).

In the last decade, observations and measurements from direct access as well as valuable insight from remote sensing and geophysical data, have helped characterise contemporary grounding line environments and the processes acting at the time of observation. At the Whillans Ice Stream grounding line, one of the best studied contemporary grounding lines, a grounding zone wedge is actively forming (Anandakrishnan et al., 2007) and processes, including channelised meltwater delivery (Horgan et al., 2013), tidally-induced compaction of till (Christianson et al., 2013) and basal melt-out of englacial debris

(Christianson et al., 2016), are thought to contribute to grounding line dynamics. Observations and modeling results demonstrate coupling between ice shelf change and grounding line movement, indicating that grounding lines are sensitive to ice shelf buttressing (e.g., Shepherd et al., 2004; Goldberg et al., 2009). Longer-term and larger-scale modeling has shown that grounding lines are sensitive to bed geometry and the presence or absence of topographic pinning points (Jamieson et al., 2012). Despite these advances, most observations of grounding line processes and, importantly, the response of the

grounding line to those processes, are limited in spatial coverage and relate to timescales of years to decades at best. A comprehensive understanding of grounding line stability and the rates, magnitudes, and timescales of change is therefore precluded.

Grounding line landforms (grounding zone wedges and terminal moraines, Fig. 1B-D) directly mark present and former grounding line positions, and represent the history of sedimentation during periods of grounding line position stability.




Sediment is transported by glacial and glaciofluvial processes to the grounding line, where it is either deposited and a landform builds, or is further transported into the marine environment by sediment plumes. Terminal moraines, here referring to any moraines that form at grounding line positions, are thought to form by a variety of sedimentation processes, including lodgement and deformation of subglacial till; pushing and squeezing of ice-marginal sediments; rockfall, dumping, and melt-out of englacial debris; as well as glaciofluvial sediment delivery and suspension settling (Powell and Alley, 1997; Batchelor and Dowdeswell, 2015). Grounding zone wedges are rather distinct, low profile landforms with an asymmetric morphology (e.g., Anderson, 1999; Anderson and Jakobsson, 2016; Batchelor and Dowdeswell, 2015). Previously described as till tongues (King et al., 1991), till deltas (Alley et al., 1987, 1989), and diamict aprons (Hambrey et al., 1991; Eittreim et al., 1995), they are composed of prograding strata of dilatant deforming till (King, 1993; Powell and Alley, 1997; Anderson, 1999; Dowdeswell and Fugelli, 2012; Batchelor and Dowdeswell, 2015; Demet et al., in review). Both types of grounding line landform have been observed to contain features described as grounding line fans: lobate or bulbous deposits building from a point source, and linked to both glaciofluvial deposition at the mouth of a subglacial channel and to remobilization of grounding line sediments by gravity flows (Powell and Alley, 1997; Bjarnadóttir et al., 2013).

It should follow that if different sets of processes build contrasting landforms, then observations of landforms can be inverted for the conditions and processes operating at palaeo-grounding lines. However, a consistent view of what fundamentally controls why one landform type is produced rather than another is lacking. The presence of an ice shelf is argued by some to be critical to the production of grounding zone wedges (e.g., King, 1993; Dowdeswell and Fugelli, 2012; Batchelor and Dowdeswell, 2015). The restricted vertical accommodation space underneath an ice shelf accounts for the low-profile wedge morphology and promotes growth by progradation; conversely, a moraine ridge can build at an ice-cliff terminus where its vertical growth is unrestricted. Powell and Alley (1997) argue that an ice shelf is not critical, but rather the subglacial thermal and hydrological regimes and their effects on the mode of sediment delivery control terminal landform development. Dilatant deforming sediment, whose production is encouraged by a meltwater system that predominantly drains through porewater (Darcian processes), has a low angle of repose and will build a low-relief wedge irrespective of accommodation space. Feedback between wedge building and grounding line stability causes the grounding line position to advance over its wedge, continuing to deform underlying sediment and, in some cases, produce subglacial lineations on the wedge topsets (e.g., Ottesen et al., 2005; Bart and Owolana, 2012; Jakobsson et al., 2012). Where meltwater is instead in high abundance and drains through a channelised system, subglacial sediments are less easily deformed. Grounding line sedimentation is dominated by non-wedge building processes, including glaciofluvial deposition, and terminal moraines and fans build with a higher angle of repose. Bjarnadóttir et al. (2013) challenge this meltwater/sediment delivery model for grounding line landforms, reporting observations of meltwater fans (channelised meltwater) within grounding zone wedges (distributed meltwater). However, in all these cases net addition of sediment to the grounding line implies that the size of an eventual landform will reflect a combination of sediment flux/accumulation and *time* and should, therefore, provide some measure of grounding line 'stability'.



Enhanced coverage and resolution of bathymetric data (e.g., multibeam sonar) acquired over the last 10-15 years from numerous continental shelf and ice sheet settings reveal vast swathes of grounding line landforms. These provide a wealth of data on grounding line retreat following the last glacial maximum, and offer an opportunity to extract information about grounding line processes and sensitivity across a range of glaciological, topographic and oceanographic settings. Here we

characterise morphological traits and the spatial distribution of 6,275 grounding line landforms from the western Ross Sea continental shelf, formerly occupied by a marine-based sector of the East Antarctic Ice Sheet, to characterise landform morphology, examine those factors that control landform morphology and distribution, and explore drivers of grounding line stability and instability.

## 2 Data and methodology

Multibeam bathymetry was collected on cruise NBP1502A aboard the RVIB Nathaniel B. Palmer using a Kongsberg EM-122 system in dual swath mode with a 1°x1° array and 12 kHz frequency, surveying with approximately 30-60% swath overlap and with regular sound velocity control. At this frequency, the system vertical resolution is on the order of centimeters and, in 600 m water depths typical of our study area, horizontal resolution is ~6 m (following Jakobsson et al., 2016). The NBP1502A data were cleaned and gridded at 20 m cell size, and combined with re-processed legacy multibeam

data from the LDEO-Columbia University Marine Geoscience Data System archive at www.marine-geo.org at a grid-cell size of 20-40 m, depending on the resolution of the original component datasets. Sub-bottom acoustic data were collected with a Knudsen chirp 3260 system during cruise NBP1502A using a frequency of 3.5 kHz and a 0.25 ms pulse length. Two-way-travel time was converted to depth using a sound velocity of 1,500 m s$^{-1}$ and to sediment thickness using velocities of 1,500-1,750 m s$^{-1}$.

Grounding line landforms were mapped based on visual identification and interpretation (Fig. 2). Morphometric parameters of individual landforms were calculated using standard line geometry tools in ArcGIS and a peak picking function in Matlab from transects across grounding line landforms (Fig. 2). We explore correspondence between morphometry, landform distribution, topography, and sediment distribution. Analyses are detailed in the Supplementary Methods.

## 3 Grounding line landform morphology

Using high-resolution multibeam bathymetry data, we mapped 6,275 grounding line landforms that visually present two distinct populations (Fig. 2): quasi-linear, closely spaced, symmetric ridges interpreted as recessional moraines (Fig. 3A-D; n=4,586); and asymmetric ridges with a smeared appearance, interpreted as grounding zone wedges (Fig. 3E-G; n=1,689). Whereas grounding zone wedges occasionally are overprinted by glacial lineations (Fig. 3E), glacial lineations are never associated with the mapped moraines (Fig. 3A-B). Exclusively amid a field of recessional moraines, we observe a group of

irregular ridges with variable amplitudes and orientations that both cut across and partially align with the moraines (Fig. 3H-I; n=189). We interpret these as basal crevasse squeeze ridges likely formed in a subglacial, yet near-grounding line, setting.



Morphological analyses show that as a population, landforms interpreted as recessional moraines are low amplitude (μ=2.0 m, SD=1.2), narrow in the cross-profile (i.e. along ice flow) direction (μ=83 m, SD=39.1), are spaced typically less than 1 km apart (μ=419 m, SD=328), tend towards a symmetric cross-profile, and have a straight form (Fig. 4A). Landforms classified as grounding zone wedges are typically higher amplitude (μ=6.2 m, SD=8.0), wider in the cross-profile direction

(μ= 522 m, SD=724), more widely spaced (μ=2,100 m, SD=3,430), asymmetric, and relatively sinuous (Fig. 4B). Among the whole population of landforms, width is found to scale with amplitude (Fig. 5A), a trait that is consistent with grounding line landforms reported from other marine-terminating ice sheet settings worldwide (Fig. 5B-C). A notable trend, however, is that western Ross Sea grounding line landforms have smaller widths and amplitudes than most landforms observed elsewhere, with western Ross Sea grounding zone wedges being the smallest documented grounding zone wedges (Fig. 5B) and western

Ross Sea recessional moraines overlapping with landforms elsewhere identified as De Geer moraines (Fig. 5A; Ojala et al., 2015; Todd et al., 2016). We additionally find that the larger the two-dimensional form of the landform, the greater asymmetry it has developed (Fig. 5D), while Fig. 5E-F illustrate that these properties are also correlated with landform sinuosity. Grounding zone wedges in general are found to be larger, more asymmetric and sinuous and, furthermore, much more variable in each of these properties compared to the tight distributions and consistent form of recessional moraines.

Individual morphometric parameters show overlapping distributions and imply a continuity of form between recessional moraines and grounding zone wedges (Fig. 4). However, a more holistic description that considers two or more landform properties (Fig. 5) tends to separate the landform population into two groups, consistent with visual interpretation of two distinct landform types. Grounding zone wedges and recessional moraines occur within clusters of numerous landforms of the same morphotype, which transition from one morphological end-member to another both laterally across a single time-

synchronous grounding line (Fig. 6A-C), and within a grounding line retreat sequence, indicating a temporal switch in landform type (Fig. 6D).

Our observations point to variability in grounding line processes and environments that can lead to a spatial (lateral) and/or temporal switch between two distinctly different landform products. We now ask: what grounding line settings or processes may control the production of contrasting landforms and what, consequently, can we learn from the style and distribution of

grounding line landforms about the (in)stability of a retreating ice sheet?

## 4 Controls on grounding line landform morphology and distribution

For marine-based ice sheets, a state of grounding is fundamentally a function of ice thickness and water depth. A range of glaciological and oceanic processes and topographic settings can affect this relationship (Fig. 1A) and, one may hypothesise, also affect the landform product of grounding. *Bed topography* has a direct control on the relationship between ice thickness

and water depth, and therefore the grounding line position. Topography also exerts an indirect control on grounding line processes, creating spatial variability in ice flow velocity and basal sediment flux, influencing tidal amplitude and near-grounding line ocean circulation, and thereby affecting both the tendency towards buoyancy and grounding zone mass





balance. *Grounding line sedimentation*, importantly, itself serves to build relief at the grounding line, which has been identified as a potential feedback on grounding line position stability (e.g., Alley et al., 2007; Christianson et al., 2016). Processes at the ice-bed interface determine basal sediment transport mechanisms and fluxes, and basal and near-grounding line sedimentary processes have therefore been considered fundamental to the production of different grounding line

landforms (e.g., Powell and Alley, 1997; Bjarnadóttir et al., 2013). Finally, the presence or absence of an *ice shelf* exerts a major control on ice flow by providing back-stress to grounded ice (Fig. 1A; e.g., Scambos et al., 2004; Fürst et al., 2016), affects mass balance at the grounding line via effects on submarine melting and on calving rate, and places a limit on sediment accommodation at the grounding line.

We use our dataset to evaluate three groups of potential controls on grounding line landform morphology: i) topographic

setting, ii) grounding line sedimentation, and iii) presence or absence of an ice shelf.

### 4.1 Topographic setting

We consider here that topographic factors including water depth, the bed slope, and regional topographic configuration could affect landform development. Firstly, grounding line landforms of both types are widely distributed across a range of water depths and bed slopes (Fig. 2, 7). Collectively, they occupy a particular window of available depths in the western Ross Sea

(Fig. 7A-C), typically within or on the flanks of well-defined glacial troughs (Fig. 2). Neither landform type occurs in limited areas of particularly shallow (<300 m) and deep (>1000 m) waters, indicating that water depth exerts a moderate control on grounding line landform construction, or rather grounding in general. The lack of landforms in shallow water depths found at bank tops could result from slower flowing or stagnant ice, as suggested by Shipp et al. (1999) and Halberstadt et al. (2016), not conducive to grounding line landform growth. In the deepest water depths, ice may not have re-

grounded during retreat or grounding line positions are not expressed as discernible landforms.

Recessional moraines and grounding zone wedges occur in a similar range of water depths, suggesting water depth alone does not dictate the formation of a particular landform (Fig. 7A-C). Furthermore, landform types display variable spatial associations with respect to water depth. We find both landforms at similar water depths (Fig. 6A), recessional moraines shallower than grounding zone wedges (Fig. 6B), and grounding zone wedges shallower than recessional moraines (Fig. 6C).

The absence of a consistent relationship between either morphotype and particular water depths again implies that water depth has little direct influence on the type of grounding line landform. We question, therefore, whether water depth has an influence on landform-building processes.

There are very weak preferences of landform type for particular bed slopes or regional topographic configuration. Collectively, grounding line landforms span the full range of bed slopes that exist in the western Ross Sea (Fig. 7D-F), but

moraines appear to favour particularly low slope beds (Fig. 7E). The lowest (i.e. flat, <0.1°) slope beds, generally found within troughs formerly occupied by ice streams, also have a more diverse range of forms (Fig. 2). This suggests that the formative controls on landform morphotypes are more variable on flat beds. Where a slope is present (>0.1°), recessional



moraines show a slight tendency for orientations perpendicular to slope contours (Fig. 8A), indicating that grounding lines expressed by these moraines were laterally grounded in a range of water depths. On the contrary, grounding zone wedges more commonly follow slope contours (Fig. 8B), and mark individual grounding line positions that were laterally situated at more-or-less equal water depths.

Since recessional moraine orientations appear to be insensitive to bed slope changes, and yet they commonly populate flat beds, we find that recessional moraine formation is not sensitive to topographic setting. Grounding zone wedges more commonly adjust orientation to slope contours and, in some cases, are observed on likely pinning points, including isolated relief on the seafloor (Fig. 8C-D) and bank slopes (Fig. 8E). However, at full population scale, landform morphologies are not distinctly associated with certain water depths or bed slopes. Our analyses, therefore, indicate that topographic control on

grounding line landform development is weak and localized, and grounding zone wedges and recessional moraines must be primarily differentiated by alternative controls or processes.

## 4.2 Grounding line sedimentation

Grounding line landforms are constructional, positive relief features, and consequently the sedimentary processes involved in the delivery of sediment and the construction of relief are important to the resultant landform morphology. The style and

magnitude of subglacial to grounding line sedimentation should influence landform growth, and the resulting form and size has often been linked to basal sediment fluxes and the duration of grounding (Powell and Alley, 1997; Batchelor and Dowdeswell, 2015; Bart et al., 2017). Implicit in this interpretation is that grounding line landforms are depositional, and that they grow with sediment input and with time. Here we first examine evidence in our dataset for the mechanisms of sedimentation.   We then assess the importance of grounding line sediment accumulation in accounting for differences

between landform types.

### 4.2.1 Sedimentation mechanism

Two styles of sediment accumulation at grounding lines can conceptually be distinguished: (1) deformation of sediments at the grounding line by push and squeeze, and (2) deposition (i.e., net input of sediment) at the grounding line supplied by mobilized subglacial sediments and release of debris from overlying ice by melt-out at or immediately in front of the

grounding line. But, do these contrasting sedimentation styles produce two distinct grounding line landform morphotypes? Morphological signs of scour and push occur at the lateral transition from a grounding zone wedge to recessional moraine along a single grounding line position (Fig. 6A, C). In Fig. 6C, recessional moraines appear to 'peel off' from grounding zone wedges, where wedge sediment appears to be pushed forward to form a narrower ridge. In these examples, there is some element of push of grounding line sediment over a distance that is comparable to grounding zone widths (~100-500 m).

The occurrence of crevasse-squeeze ridges within a recessional moraine field (Fig. 3H) is evidence for squeeze of subglacial sediment upward into vacant space at the ice base, but the considerably greater amplitude of the crevasse-squeeze ridges (Fig. 3I) leads us to question the extent to which this same process occurs at the grounding line. While recessional moraines



appear to form on flat surfaces, generally free of excavated lows that could be obvious source areas of pushed sediment (Fig. 3A, B, 6), sub-bottom acoustic profiles only occasionally show deposition onto a preserved lower surface represented by an acoustic reflection horizon (Fig. 9A). More commonly, recessional moraines are formed *from* the upper sediment unit, which would suggest that relief has been created by local sediment deformation. However, there is no strong evidence in our

datasets for determining whether the moraines formed from cessation of along-flow transport of a deforming layer (an input flux), or from localized bulldozing of existing sediments. Any input flux would need to be extremely consistent both spatially and temporally to produce the consistent, quasi-linear and symmetric moraine morphology observed.

Grounding zone wedges have been widely shown to be depositional products that accumulate by progradation of sediments delivered to the grounding line (e.g., Anderson, 1999; Batchelor and Dowdeswell, 2015). Sediment delivery to the

grounding line is assumed to be primarily from a conveyer belt of deforming till at the base of the ice sheet. This material is subsequently transported down the foreset slope of the wedge by sediment mass movement (Demet et al., in review). Here, the asymmetric morphology and distinct stoss-lee slope transitions of grounding zone wedges (Fig. 3E-F) are consistent with landform topset aggradation and foreset progradation, although individual topset and internal foreset beds are not resolved in our high-frequency acoustic data from these relatively small landforms, contrary to lower-frequency seismic records of

larger documented grounding zone wedges (Fig. 1D; e.g., Batchelor and Dowdeswell, 2015). Undisturbed buried horizons beneath a variety of grounding zone wedges in our dataset (Fig. 9B-C, 10B) clearly indicate that wedge relief is due to the addition of material at the grounding line. Landform arrangements further reveal that wedges have prograded over older recessional moraines and grounding zone wedges (Fig. 3E, G). Grounding zone wedge profiles are remarkably consistent in their general shape, with short and relatively steep distal slopes; we do not observe any lobate or bulbous fan deposits along

wedge fronts (e.g., McMullen et al., 2006; Bjarnadóttir et al., 2013; Fig. 1C) that would indicate point sources of sediment such as meltwater conduits and/or remobilization of sediments via sediment gravity flows. However, earlier collected side-scan sonar data document small-scale slumping on the foreset of a grounding zone wedge in southern JOIDES Trough (Fig. 11), perhaps indicating the delivery of relatively cohesive sediment to the grounding line conducive to viscous sediment gravity flows. Thus, our dataset shows that the processes responsible for delivering sediment to the grounding line are

spatially rather uniform, although they may differ in rate and flux.

Acoustic profiles of some smaller grounding zone wedges show signs of active deformation through ~10 meters of sediment thickness, at and behind the grounding line, which has destroyed a buried acoustic horizon that is seen immediately distal to the group of wedges (Fig. 9D). Folded foreset toes and streamlined subglacial lineations that overprint grounding zone wedge topsets further indicate that ice actively molds (deforms) the bed as it builds the wedge and then holds this grounding

zone position. The association with lineations also reveals that grounded ice has locally advanced over its own sediments; a depositional model of progradation is in these cases accompanied by shallow subglacial sediment deformation at the ice-bed interface.

Our data do not support one distinct sedimentation mechanism being responsible for one distinct morphotype. Recessional moraines show positive evidence for consistency in size and form, and therefore sedimentation mechanism, that is consistent



across hundreds of meters of the grounding line (Fig. 5). Grounding zone wedges show clear examples of net accumulation of sediments over a pre-existing surface, and of active deformation; wedges that display evidence of sediment progradation and deformed, lineated upper surfaces (Fig. 3E) suggest that both deformation and deposition can be jointly responsible for landform construction.

## 4.2.2 Sediment flux and duration

Landform width and amplitude are positively correlated (Fig. 5A-C), in the case of both recessional moraines and grounding zone wedges. At the smallest end of the global population, the tight morphological clustering of moraines in the western Ross Sea may suggest that there is a limit imposed on their eventual size. Such a limit must be either inherent to their process of relief creation or due to a limited net input of sediment due to low delivery flux and/or occupation time of a grounding position. Increases in width and amplitude characterise grounding zone wedges, in which landform asymmetry and sinuosity additionally develop with increasing size of the landform (Fig. 5D-F). These relationships suggest that grounding zone wedges grow as a function of sediment supply over time, and that variability in accumulation in both space and time will yield variable morphologies with heightened sinuosity developing with landform growth (Howat and Domack, 2003). Since growth is inherently a function of both sediment availability and time, these properties can be difficult to disentangle. Does a larger grounding line landform represent more time or a greater basal sediment flux?

A paired group of grounding zone wedges and recessional moraines allows us to isolate the time factor of sediment accumulation, where grounding zone wedges laterally transition to recessional moraines (Fig. 6A). We select a sequence of these landforms that is bounded by a laterally continuous grounding line at both a distal and retreated position, each representing a time-synchronous grounding line position (Fig. 10A). In this group, an individual grounding zone wedge has an average cross-section (i.e. sediment content) 8.3 times larger than that of an individual moraine. The full assemblage of retreating grounding zone wedges has 4.55 times more sediment (in cross-profile) than the neighboring assemblage of moraines, while there are twice as many individual moraines. Therefore, in two parallel corridors, the sediment flux at each grounding line position is higher in the grounding zone wedge group *and* the occupation of each individual grounding line position must be longer during grounding zone wedge construction. Based on these observations and the general consistency of recessional moraine size across the western Ross Sea, we suggest that lower sediment flux and time are both factors that limit moraine growth.

In the above example we find a difference in sediment flux to contrasting landform types. Within a grounding zone wedge in southern JOIDES Trough we find spatial variability in sediment thickness within a single landform. Sub-bottom acoustic data detect a buried surface beneath the acoustically transparent grounding zone wedge sediment unit, enabling us to map the spatial distribution of sediment accumulation on top of the underlying (i.e. older) substrate (Fig. 10B). We find that the sediment thickness at the grounding zone wedge front is laterally variable, with peak thickness in the centre-west (μ=12.9 m up to 2 km behind topset-foreset break), thickness minima to the far west and in the eastern lobe (c. 2.4 m), and moderate





thickness on the eastern flank (c. 6.6 m) and within a pronounced embayment (c. 8.5 m). In the along-flow direction, the grounding zone wedge unit also thickens and thins towards a maximum distal thickness.

Variable sediment thickness within a single grounding zone wedge points to differences in sediment flux to the grounding line. A variable sediment flux, over a scale of 100s of meters to several kilometres, may be linked to factors such as sediment

delivery from a contrasting source (different grain size, porosity, rheology), the basal thermal regime and hydrology, and any glaciological properties (ice thickness, surface slope, ice composition) that in turn affect these factors. In several groups of grounding zone wedges in our dataset, we observe embayments in the wedge front that contain channels (Fig. 12; Simkins et al., 2017). This suggests a link between the position of subglacial meltwater channels, and the development of sinuosity (creation of embayments) in the grounding line. Contrary to cases where subglacial conduits are thought to provide point

sources of fan sedimentation at a grounding line (Powell and Alley, 1997; McMullen et al., 2006; Bjarnadóttir et al., 2013), here we observe reduced availability of sediment at the grounding line associated with basal channels. We hypothesise this may result from non-deposition of sediment at the grounding line due to enhanced transport by glaciofluvial processes or, alternatively, embayments may be due to porewater drainage by the channel, sediment stiffening and reduced subglacial transport by deformation processes (e.g., Christianson et al., 2013). In the latter case, we would expect excessive thickening

of sediment behind a grounding zone wedge embayment. Within the coverage of our data, we observe along-flow thickening to the wedge front (Fig. 10B) but to a *lesser* degree within the embayment than to the side, indicating no excess of sediment accumulation around the channel or embayment. Furthermore, the occurrence of meltwater plume deposits in cores seaward of palaeo-grounding line positions (Simkins et al., 2017; Prothro et al., 2018) supports flushing of sediment through the grounding line by basal meltwater, producing lateral variability in the magnitude of landform progradation and resulting in

highly sinuous grounding zone wedges.

Development of sinuosity via spatially reduced or enhanced deposition is a function of variable sediment supply and of time. A longer duration of standstill will permit the variability in transport and deposition rates to enhance the sinuosity of the eventual form. The paired sequence shows that *both* flux and occupation time are greater in the case of grounding zone wedges than moraines, while the stacking of grounding zone wedges (Fig. 6C) and their association with sites of topographic

pinning also imply greater construction time. The progressive development of traits such as asymmetry and sinuosity, both atypical of moraines, may indicate that low-amplitude recessional moraines are the proto-feature with limited construction time and limited supply. This would mean that as either/both of these increase, grounding zone wedges preferentially develop. This model is difficult to reconcile with examples of much larger individual terminal moraines globally (Fig. 5C). Nonetheless, we show here that sediment flux and time are important controls on landform type and appear to control the

development of landform asymmetry and sinuosity.

### 4.3 Presence or absence of an ice shelf

Ice shelf presence/absence has been postulated as an explanation for contrasting grounding line landforms, where accommodation space at the grounding line is limited under an ice shelf and promotes low relief, asymmetric grounding




zone wedge development, while an ice cliff has unlimited accommodation and a symmetric moraine can build upward (Powell, 1990; Dowdeswell and Fugelli, 2012; Batchelor and Dowdeswell, 2015). The existence of two end-member landform types is tempting to explain by a mechanism with two equivalent end-member states. If grounding line landform morphology could clearly be associated with ice shelf configuration, then we could use the presence of landform type as a proxy for palaeo-ice shelf presence/absence and identify grounding lines that could have been influenced by ice shelf back-stresses.

Our data show that grounding zone wedges in the western Ross Sea have a higher amplitude than recessional moraines (Fig. 5). The condition for the argument given above is therefore not upheld or, at least, an additional process or factor is required to account for inhibited moraine growth. Furthermore, we might expect topographic highs to maintain grounded ice whilst ice over deeper troughs would tend towards flotation and preferentially form an ice shelf, as is argued for late stage deglaciation in the western Ross Sea (Yokoyama et al., 2016). However, there is not a consistent relationship between recessional moraines and grounding zone wedges and their topographic context (e.g. Fig. 7B-C) that would support this association. Transitions between the two landform types along a single continuous grounding line (Fig. 7A) in a comparable topographic setting are also not straightforward to reconcile with an ice shelf/no ice shelf condition.

The presence of deep iceberg furrows on the continental shelf provides compelling evidence for calving at or near the grounding line (Anderson, 1999; Jakobsson et al., 2011; Yokoyama et al., 2016), while Wise et al. (2017) argue that deep iceberg furrows in Pine Island Bay record episodes of ice-cliff instability. In western Ross Sea, such furrows formed on the outer continental shelf, and are followed in the retreat sequence by closely spaced recessional moraines (Halberstadt et al., 2016). However, fields of deep iceberg furrows are lacking in association with the majority of mapped grounding line landforms discussed here. If deep iceberg furrows mark an ice cliff setting at the grounding line, does the absence of deep iceberg furrows imply the presence of an ice shelf? We cannot yet convincingly attribute grounding line landform type to the presence or absence of an ice shelf.

## 4.4 Discussion of controls on landform morphology

At a regional, trough-wide scale, the production of grounding line landforms and their particular morphologies do not appear to be fundamentally controlled by properties of the bed topography such as water depth and bed slope. Locally, the presence of topographic relief (banks, seamounts) encourages the construction of grounding zone wedges, flat (<0.1°) beds are characterised by a heightened range of grounding line landform morphologies, and grounding zone wedges adjust to the local bed slope direction. Recessional moraine orientations independent of slope direction might suggest that they were more likely formed at an ice cliff than under an ice shelf, since shelf formation is fundamentally determined by the water depth relation to ice thickness and the grounding line supplying an ice shelf would therefore more likely follow the local bed shape. However, although an intuitive hypothesis, the presence or absence of an ice shelf cannot fully explain either grounding line landform morphology or distribution in the western Ross Sea. The overlapping size range of grounding zone





wedges and moraines from other deglaciated margins (Fig. 5B-C) further suggests that ice shelf presence/absence, controlling grounding line accommodation space, cannot be the only control on landform morphotype occurrence.

Whether the relief of recessional moraines and grounding zone wedges is created by local push and squeeze or by deposition from sub/englacial transport is inconclusive, but some combination of sediment supply and time is clearly important to the

eventual form. Landform growth that begins due to grounding line deformation and transitions (either spatially or temporally) to landform growth by deposition would explain why there is more evidence of deformation in the development of relatively small moraines (that might seed grounding zone wedges) and deposition for larger, more variable grounding zone wedges. However, terminal moraines in other locations are more variable in size (Fig. 5C) and span the full range of documented grounding zone wedges (Fig. 5D); therefore, our dataset of thousands of small recessional moraines in the

western Ross Sea (and similar scale De Geer moraines elsewhere) are either genetically different to larger documented moraines, or perhaps characteristic of short-lived grounding line positions.

There are many processes that vary across a continuum that all likely influence grounding line configuration and sedimentation, so why do we not observe morphological products that also vary across a continuum? Rather, we observed a binary product – either grounding zone wedge or recessional moraine. Individual morphological/spatial landform

characteristics may overlap but in combination they produce two species that are visually very distinct from each other. A mechanism that itself is binary is an appealing way in which to explain a set of products that is binary. However, our data do not offer such a solution, and instead, several factors offer partial yet inconclusive explanations of landform morphology. Of these, time appears to be important to eventual grounding zone wedge morphology, and it therefore follows that these landforms hold some information about the stability (duration) of grounding line positions. In the next section we explore to

what degree landform morphology and landform distribution may lead us to interpret aspects of grounding line (in)stability.

## 5 Implications for grounding line (in)stability

Grounding line 'stability' can be conceptualized in numerous ways. One possible definition may be in terms of grounding line *sensitivity to change* in position, that could be expressed as a likelihood of a major or minor grounding line response to certain forcings. Alternatively, stability could be defined by the *duration of grounding line position* occupation, or the

*magnitude of the retreat event* when a grounding line vacates a former position. Stability could also be expressed as *consistency (or predictability)* in grounding line position duration, or consistency of the magnitude of retreat event across numerous back-steps. Given these different facets of the concept of stability, would consistently small (large) magnitude retreat events punctuated by short (long) periods of grounding line position occupation reflect stable or unstable grounding line behaviour? Here we consider 'stability' to be twofold: duration (and consistency of duration) of grounding line

positions, and magnitude (and consistency of magnitude) of retreat events. To address these definitions of stability, we next discuss retreat patterns based on the distribution of grounding line landforms in the western Ross Sea and what landform morphology indicates about landform construction feedbacks on stability.




## 5.1 Stable or unstable retreat?

Retreat sequences defined by recessional moraines indicate short-distance retreat steps (mean spacing μ = 419 m) whose magnitude is extremely regular (SD = 328 m; Fig. 4). Grounding zone wedges, on the other hand, are more widely (μ = 2,100 m) and less consistently spaced (SD = 3,430 m), just as they are less consistent in their overall form. These population data are consistent with the extremely regular visual appearance of moraine sequences comprising 10s-100s of individuals (Fig. 3A, B), and with much more varied examples of grounding zone wedge retreat assemblages in which there may be either a noticeable gap between the toe of one feature and the proximal slope of the next feature (Fig. 6A) or pronounced stacking of individual features (Figs. 3E, 6B, 6C). In the eastern Ross Sea, even larger (>100 km) magnitude back-steps are separated by extensive zones of pristine mega-scale glacial lineations, recording significant retreat events when ice floated off the bed in the intermediate area, thus preserving the underlying subglacial landform assemblage (Mosola and Anderson, 2006; Bart and Owolana, 2012; Halberstadt et al., 2016; Bart et al., 2017). Overall, grounding lines that produce a grounding zone wedge undergo retreat in a much more inconsistent manner than those favouring moraine formation, with the magnitude of retreat events being more variable where clusters of grounding zone wedges form.

As with their spacing, recessional moraines have a tight size distribution (Fig. 4, 5), indicating not only consistency in retreat event magnitude but also in the duration that a grounding line position is occupied. Their small size would suggest that this duration is typically short, following our finding that a paired sequence of laterally continuous wedges and moraines (Fig. 6A) has both lower sediment supply and shorter occupation time where the grounding line produces a moraine. Features of comparable scale to our Ross Sea moraine population (e.g. Fig. 5C) are commonly interpreted as De Geer moraines, considered to form annually or sub-/multi-annually (Lindén and Möller, 2005; Todd et al, 2007; Ojala et al., 2015). Grounding zone wedges represent longer duration grounding line positions, indicated by both their larger sediment content and, we argue here, by their greater sinuosity and asymmetry that both scale with landform size and take time to develop. Published estimates of grounding zone wedge formation time suggest timescales of decades to millennia (Anandakrishnan et al., 2007; Nygård et al, 2007; Jakobsson et al., 2012; Klages et al., 2014; Bart et al., 2017), though these typically relate to individuals larger than those found within our dataset (Fig. 5B). Our paired group of small-scale (<10 m in amplitude) grounding zone wedges and recessional moraines indicates grounding zone wedge formation timescales approximately twice as long as their moraine counterparts. The wedges in this group are among the smallest in our dataset and we estimate grounding zone wedge occupation in our study area on multi-annual to centennial timescales, as suggested by Simkins et al. (2017). Therefore, grounding zone wedges represent longer duration standstills, indicating a more prolonged grounding line configuration. Contrastingly, recessional moraines indicate a mode of retreat that is regular and consistent (i.e. predictable), yet more frequent and punctuated by short-lived grounding positions. These landform-based patterns of retreat are conceptually summarized in Fig. 13A-B.

## 5.2 Drivers of retreat



Much of our landform population in the western Ross Sea comprises individuals arranged in groups of alike morphotypes (Fig. 2, 3, 7). While the frequency of retreat events from individual landform to landform is on timescales of years to centuries, this clustering of distinct end-member landform types indicates that (i) the timescale for a change in process or grounding line setting that would yield a different type of product is extremely abrupt, based on the lack of transitional

landform types, and (ii) that once the formational process/environment has changed, it is maintained for a duration significantly longer than the construction time for a single landform.

Retreat of a grounding line must be fundamentally driven by a change to the buoyancy condition, that causes ice at the grounding line to lift off from one position, or a rate of grounding line ablation, via melt or calving, that exceeds the incoming grounding line ice flux, which may drive a change in the buoyancy condition (Fig. 1A). The regularity of moraine

sequences suggests a cyclic process that would produce short-lived grounding but controlled and small-scale retreat magnitude (Fig. 13B). This retreat style is most likely driven by changes in ablation (mass balance) conditions. Possible mechanisms for cyclic control on grounding line retreat could include annual/multi-annual sea ice variability that has been linked to reduced calving and alters continental shelf ocean circulation (Hellmer et al., 2012), climatic phenomena like El Niño Southern Oscillation which can alter ice shelf mass balance (Paolo et al., 2018), tidal cycles causing sufficient calving

and/or basal melting to drive grounding line retreat (Jakobsson et al., 2011), or regularly paced subglacial meltwater drainage events that could cause plume-driven melting (Le Brocq et al., 2013; Alley et al., 2016). Collectively for contemporary Antarctic ice shelves, basal melting accounts for over 50% of ice shelf mass balance loss (Depoorter et al., 2013; Rignot et al., 2013), concentrated near the grounding line (e.g. Jenkins and Doake, 1991; Rignot and Jacobs, 2002); calving accounts for the other half. Since a number of processes that operate on a regular/cyclic basis can control ablation at grounding lines,

we suggest that these same processes could also contribute to consistently small-scale retreat events, revealed by small, closely-spaced recessional moraines in the western Ross Sea.

Sequences of clustered moraines would suggest that these processes, in and of themselves, are not enough to trigger exceptional large-magnitude - one may argue 'unstable' - retreat events, but rather produce steady, controlled retreat. Recessional moraine sequences are rarely terminated by a large-magnitude retreat event, which could argue in favour of an

unstable threshold response to prolonged small-scale ablation forcing, but rather moraines switch to a grounding zone wedge. A short duration at each grounding line position may nonetheless mean the removal of a large volume of ice in a relatively short period of time through steady processes. This is in contrast to grounding zone wedges that mark longer grounding line position occupation, yet larger and less consistent retreat events. So what controls retreat assemblages that are not controlled by cyclic processes?

Even though it is likely that the cyclic processes that take place at grounding lines expressed as moraines are also ongoing at positions marked by grounding zone wedges, sensitivity to processes occurring at more-or-less regular intervals appears to be reduced where grounding zone wedges are present. This suggests that the grounding line is buffered from processes that drive short-term (annual/multi-annual) variability in ablation/buoyancy. Such a buffer could be due to i) a long-term shift in ocean access to the grounding line (e.g. circulation change, change in ice shelf and/or sub-ice shelf cavity geometry) such





that calving and basal melt rates fall below a threshold (relative to ice flux) for enacting grounding line change; ii) an increase in the ice thickness to water depth relation (reduced buoyancy), such that the grounding line ice flux or sediment flux to the grounding line overrides any small-scale variability in ablation rates via calving or basal melt; or iii) a feedback with the processes of wedge construction itself. A fundamental change in the ice thickness - water depth relation away from

floatation might desensitize grounding lines to terminal ablation processes and promote longer occupation of a grounding line position. Grounding zone wedges on topographic pinning points reflect this condition; the slight tendency of grounding zone wedges to adjust their orientation and shape with respect to local bed slope suggests these locations may be sites of pinning but are sensitive to buoyancy control (e.g., Fig. 8D-F). Further changes in the ice thickness-water depth relationship could destabilise the grounding position and trigger larger retreat events. Where grounding zone wedges are not associated

with pinning but rather occur at the same water depths and bed slopes as recessional moraines, we envisage two explanations for the greater occupation time of the wedge positions: (i) a function of a local difference in ablation rate/ice flux, rather than buoyancy-driven retreat; or (ii) a feedback between construction of sedimentary relief, which is largely controlled by sediment flux to the grounding line, and enhanced grounding line stability.

## 5.3 Landform feedbacks on grounding lines

Grounding zone wedges in the western Ross Sea are both longer-lived and show signs of local ice advance compared to recessional moraines. Sediment aggradation and progradation at the grounding line is accompanied in several cases by topset development of subglacial lineations. These observations suggest that grounding zone wedges stabilise grounding lines and even allow for ice advance, but that recessional moraines have no feedback on grounding line stability. The ability for grounded ice to advance during grounding zone wedge construction suggests that ice is at/near buoyancy limits and highly

sensitive to relatively small (meter-scale) changes in the ice thickness-water depth relation. As sediment is added to the landform, the depth of the seabed relative to ice thickness is reduced and allows the position of grounding to advance. Ultimately, however, retreat events from such 'stabilised' positions tend to be large (Fig. 4). The earlier sensitivity to small changes in the buoyancy relation does not manifest as incremental retreat steps.

Although grounding zone wedge growth initially encourages prolonged occupation of grounding positions and promotes

local ice advance by elevating the bed, does it also promote greater instability in the context of the magnitude of retreat events? Larger retreat events associated with grounding zone wedges suggest a threshold of stability is reached that causes inherent instability. We interpret grounding zone wedge asymmetry and sinuosity as signatures of both stabilising and destabilising feedbacks, respectively, that develop with landform growth (Fig. 5D-F, 13C). Asymmetry is a morphological expression of ice advance due to landform aggradation and progradation, and therefore reflects the stabilising aspect of

grounding zone wedges. We argue that the development of sinuosity, on the other hand, leads to a threshold of maximum stability and grounding line retreat. Several processes could lead to grounding line destabilisation associated with sinuosity, including: (i) increased contact of the ice front with ocean water, which could lead to increased melting; (ii) channelised meltwater drainage at grounding lines, which is associated with the development of embayments and the release of



meltwater plumes that contribute to melting of the ice front and ice shelf (if present) and/or increased tidal pumping; and (iii) laterally variable stresses that might produce localized shear zones or reduce lateral drag that could promote enhanced calving/crevassing potential. Such processes may promote and reinforce highly spatially variable ablation, creating sinuosity (embayments) in the larger grounding zone wedges that far exceeds the spatial scale of retreat steps associated with smaller

landforms, and potentially creating lasting change to the structure of the grounding line such that eventual destabilisation of the grounding position is larger and less predictable than in the case of smaller-scale, more ordered retreat.

Unlike grounding zone wedge growth that can both stabilise *and* destabilise grounding lines, grounding lines expressed as recessional moraines are not clearly influenced by landform presence/growth. This leads us to conclude that processes driving retreat from moraines should be independent of grounding line sedimentation. Implicit in the above is that grounding

lines producing moraines and those producing grounding zone wedges have different sensitivity to processes that trigger grounding line retreat.

## 6 Conclusions

Grounding line landforms have the potential to inform us of the processes governing the stability and retreat of palaeo-ice sheet grounding lines. From a large dataset of mapped grounding line landforms, individual morphometric properties

indicate a continuum of form. However, multi-parameter analyses support a visual classification of a binary landform product, that expresses lateral (i.e. along a single grounding line) and temporal (i.e. within a retreat sequence) transitions between clusters of two end-member morphotypes: moraines and grounding zone wedges. It is an appealing idea that a different set of controls and/or processes should dictate the formation of two different landform types. Yet, of the potential controls on landform morphology that we have explored here (topographic setting, grounding line sedimentation, and

presence or absence of an ice shelf), we find inconclusive evidence that a distinct set of controls/processes can wholly explain the formation of either morphotype.

Landform morphotype is not fundamentally controlled by water depth or bed slope, although grounding zone wedges are observed on isolated pinning points likely associated with locations of enhanced grounding line position stability. Neither can the presence or absence of an ice shelf be convincingly demonstrated to control the type of landform that results.

Inconsistent spatial arrangements of moraines and grounding zone wedges with respect to topography are difficult to reconcile with plausible ice shelf/ice cliff configurations, and the greater amplitude of grounding zone wedges than moraines suggests vertical accommodation space does not dictate landform morphology. This argument does not reject an ice shelf/cliff control, but additional factors are required to limit moraine growth in this setting.

We find that both sediment supply to the grounding line and the duration of grounding line position occupation are

important, most notably expressed in cases where grounding zone wedges laterally transition to recessional moraines: grounding zone wedges represent both a higher basal sediment flux and a longer duration of grounding than do recessional moraines. This is consistent with the development of landform shape (asymmetry and sinuosity) with the size of the

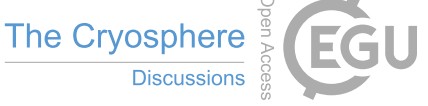

landform. A tempting conclusion is that given sufficient time and supply, a moraine would seed and develop into a wedge. This remains, however, difficult to reconcile with larger terminal moraines in other glaciated settings.

With this large dataset of morphological features associated with palaeo-grounding lines that progress 10s-100s km in the retreat direction, we are able to explore what landforms reveal about grounding line stability. Recessional moraines are

associated with short-lived grounding line positions yet record steady, small magnitude retreat events. This suggests that a regular process drives grounding line retreat, linked to steady and cyclic net loss of mass. While grounding zone wedges represent longer periods of position stability, the magnitude of retreat events is larger and more variable. Reduced ablation or a grounding line buffered against cyclic ablation processes may prolong grounding line occupation, while sediment aggradation and progradation in wedge growth may independently enhance grounding line stability. This stable phase is

reflected as asymmetry in landform morphology and in lineations on the wedge topset. However, some threshold of stability is reached to result in large 'unstable' retreat events. The development of landform sinuosity due to spatial variability in sediment transport to and deposition rates at grounding lines could potentially destabilise otherwise 'stable' grounding lines. In this regard, channelised meltwater delivery to the grounding line, ice sheet-shelf configuration, and the access of ocean water to the grounding line are likely of fundamental importance in governing grounding line shape and, therefore, ultimate

stability.

Grounding line retreat in the western Ross Sea is characterised by either i) short-lived grounding line positions that back-step with small magnitude retreat events, or ii) longer duration grounding line positions followed by major destabilisation in the form of larger magnitude retreat events (Fig. 13A-B). These contrasting behaviours vary abruptly in space and time, yet neither can be explicitly characterised as 'slow' or 'fast' retreat, nor can a single descriptor as 'stable' or 'unstable' be

applied without further qualification. 'Stability' may be conceptualized in numerous and sometimes contradictory ways. Bart et al. (2017) describe prolonged grounding line occupation and large magnitude retreat as a paradox; here we find this is a common trait of grounding line behaviour. Given non-uniform ablation and non-uniform sediment supply to a prograding landform at the grounding line, an ice margin may become increasingly prone to "unstable" (large magnitude) retreat. This study highlights the importance of understanding thresholds – potentially in the grounding line sedimentation system itself –

which may destabilise a system from an apparent state of stability, and of controls on grounding line dynamics on short (annual) to long (centennial to millennial) scales in order to project future changes in ice sheet mass balance.

**Author contribution**

L.S. and S.G. conceived the project and ran analyses. L.S., S.G., and J.A wrote the manuscript.

**Competing interests**

Authors declare no competing interests.

**Acknowledgements**



The authors thank the crew and science support personnel aboard cruise NBP1502A, as well as students from Rice University, the University of Houston, Louisiana State University, and the University of Silesia for assisting in cruise data collection. Special thanks go to L. Prothro, who provided an early draft of the grounding line retreat style schematic. This project was supported by the National Science Foundation (NSF-PLR 1246353, J.B.A.) and the Swedish Research Council

(D0567301, S.L.G.).

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





**Figure 1:** A) Schematic of a marine-based grounding line environment and processes that can influence grounding line behaviour. The minimum ice thickness ($H_{ice\ min.}$) needed for grounding is a function of water density ($\rho_{seawater}$), ice density ($\rho_{ice}$) and water depth ($H_{water}$). Examples of grounding line landforms shown in B-D. B) Regularly spaced, small amplitude recessional moraines (De Geer moraines) on the Atlantic Canadian continental shelf (modified from Shaw et al., 2009). C) Grounding zone wedges in Kveithola Trough, western Barents Sea overprinted by iceberg furrows and deposited on top of glacial lineations (modified from Rebesco et al., 2011). D) Dip-oriented acoustic profile across a grounding zone wedge in the Canadian Beaufort Sea showing prograding foreset beds that downlap on underlying surfaces (modified from Batchelor et al., 2014).

**Figure 2.** Mapped distribution of grounding line landforms in the western Ross Sea, Antarctica. Landforms include recessional moraines, grounding zone wedges and an isolated field of crevasse squeeze ridges. The landforms predominantly occur within palaeo-glacial troughs and basins: northern Drygalski Trough (NDT), southern Drygalski Trough (SDT), McMurdo Sound (MS), JOIDES Trough (JT), Pennell Trough (PT) and Central Basin (CB). Transect profiles used in morphometric analyses (**Fig. 4**) are shown by red lines.

**Figure 3.** Recessional moraines A) on relatively flat seafloor in JOIDES Trough overprint a previously active subglacial channel and **B)** on a reverse bed in Pennell Trough. **C-D)** Corresponding profiles across the fields of recessional moraines shown in **A** and **B**. **E-F)** A suite of various sizes of grounding zone wedges in Pennell Trough, most of which are overprinted by glacial lineations. The largest grounding zone wedge formed at the Last Glacial Maximum, marking the seaward-most extent of grounded ice, whereas the smaller grounding zone wedges formed during retreat across a normal sloping bed. **G)** A grounding zone wedge in JOIDES Trough with a large embayment, from which an earlier subglacial meltwater channel emanates and into which a small channel on the wedge topset leads. This grounding zone wedge is not lineated like the examples shown in **E**. **H-I)** Crevasse squeeze ridges with irregular form and variable amplitudes amid a field of recessional moraines.

**Figure 4.** Normalised frequency distribution of morphometric parameters for the population of western Ross Sea recessional moraines (n=4,586) and grounding zone wedges (n=1,689). Amplitude is equivalent to maximum landform height, width represents distance in the along-flow direction, spacing was measured as the distance between landform peaks in a retreat sequence, asymmetry was measured based on position of landform peak relative to the landform width midpoint, and sinuosity was measured across the entire mappable length of individual landforms.

**Figure 5.** Paired variable plots of landform morphometry typically distinguish grounding zone wedges from recessional moraines. **A-C)** A scaling relationship exists between landform width (distance in the along-flow direction) and amplitude in the western Ross Sea landform population, consistent with global examples of grounding zone wedges (**B**) and terminal moraines (**C**). **D)** Landform asymmetry and (**E**) sinuosity cluster differently between landform type with respect to landform size (cross-section profile area). **F)** Asymmetry against sinuosity shows clustering of low sinuosity, relatively symmetric recessional moraines. Literature sources for data in plots **B** and **C** are listed in Supplementary Material.

**Figure 6.** Recessional moraines and grounding zone wedges transition from clusters of one type to the other spatially (laterally: **A, B, C**) and temporally (in retreat sequence: **D**). Transitions occur across a variety of topographic settings (troughs: **A, D** and slopes: **B, C**), with contrasting water depth / landform type relationships, and show contrasting arrangement of individuals within a cluster, from well-spaced individuals (**A**) to stacked, overlapping wedges (**C**).

**Figure 7.** Water depth distribution across the western Ross Sea (**A**) and the water depths at which (**B**) recessional moraines and (**C**) grounding zone wedges occur. Bed slope distribution in the western Ross Sea (**D**) and the bed slope on which (**E**) recessional moraines and (**F**) grounding zone wedges are observed.

**Figure 8.** Landform aspect with respect to bed slope for (**A**) recessional moraines and (**B**) grounding zone wedges, in which 0° denotes a landform whose long axis is oriented perpendicular to slope contours ('downslope') and 90° denotes a landform oriented parallel to slope contours ('across slope'). Grounding zone wedges preferentially form on isolated bedrock highs (**C, D**) and bank slopes (**E**).

**Figure 9.** Sub-bottom acoustic data across (**A**) recessional moraines and (**B**) grounding zone wedges, where moraines appear to be formed both within the upper unit and above a reflector, and grounding zone wedges are deposited on a faint underlying reflector. (**C**) A large grounding zone wedge that clearly forms above a reflector and prograded over older strata with younger, smaller grounding zone wedges also showing signs of progradation. (**D**) Stacked grounding zone wedges appear to have disturbed a buried horizon, implying intense deformation of sediments.



**Figure 10. (A)** Paired suite of laterally continuous grounding zone wedges and recessional moraines representing retreat of a single grounding line. Increased number of moraines relative to grounding zone wedges indicates that moraines mark shorter-lived grounding line positions. Individual grounding zone wedges have an average sediment content 8.3 times larger than their moraine counterparts. **(B)** Sediment thickness of an intermediate-sized grounding zone wedge (partially visible at the bottom of **A**) that was deposited above a buried horizon (inset). Variations in sediment thickness are largest near the topset-foreset break with no distinct correspondence to the lobes and embayments of the grounding zone wedge.

**Figure 11.** Side-scan sonar image from cruise NBP95-01 across a grounding zone wedge in southern JOIDES Trough, showing slumps on the grounding zone wedge foreset surface.

**Figure 12. (A-B)** Examples of the corresponding locations of subglacial channels and grounding zone wedge embayments, resulting in sinuous grounding line configurations. Additional examples of grounding zone wedge sinuosity **(C)** downstream of and **(D)** adjacent to a large channel system.

**Figure 13. (A)** Suite of back-stepping grounding zone wedges with highly variable form and spacing some with lineated topsets and embayments, represent longer durations of grounding position stability while the retreat events are larger and less predictable. **(B)** Grounding line retreat marked by a series of recessional moraines with consistently small morphologies and relatively close spacing, indicating shorter durations of grounding position stability and smaller, yet higher frequency, retreat events. Neither contrasting style of retreat inherently indicates net rate of retreat and, thus, grounding lines forming grounding zone wedges versus moraines do not stipulate relatively fast or slow retreat. **(C)** Grounding zone wedge asymmetry and sinuosity lead to both stability and instability of grounding lines. With time (and/or increased sediment supply to the grounding line) a symmetric recessional moraine seeds an asymmetric grounding zone wedge that facilitates ice advance (a period of stable grounding). However, as asymmetry is developed, so too is sinuosity (arrows denote grounding zone wedge crestlines), which promotes increased grounding line exposure to ablation processes. Examples of multibeam bathymetry are shown in each panel and acoustic sub-bottom profile is shown for panel C.




## Fig.1                                                                p. 25

**A**

Ice flow direction

Buttressing ice flow

Ice shelf presence/absence

Sea level

Sea ice

Floating ice shelf

Upstream controls
Ice thickness & surface profile

$H_{water}$  $H_{ice\ min.}$

Grounded ice sheet

Tidal flexure

Grounding line

Tidal pumping

Basal melting

Meltwater drainage
Sediment mobility

Warm ocean water

Grounding line landform

$H_{ice\ min.} = (\rho_{seawater}/\rho_{ice})H_{water}$

Bed geology & slope

Subglacial topography

**B**

Water depth
82 m
110 m

0  0.5  1        2
km

**C**

0  2.5  5      10
km

Fan on foreset

GZW

GZW

Fan on foreset

Iceberg furrows
on topset

Iceberg furrows
on topset

Water depth
129 m
874 m

**D**

Dipping foresets

Downlap

5 km

50 m



# Fig. 2



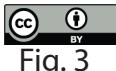

## Fig. 3





Fig. 4

A

**Recessional moraines**

B

**Grounding zone wedges**



Fig. 5



## Fig. 6





# Fig. 7





# Fig. 8





Fig. 9

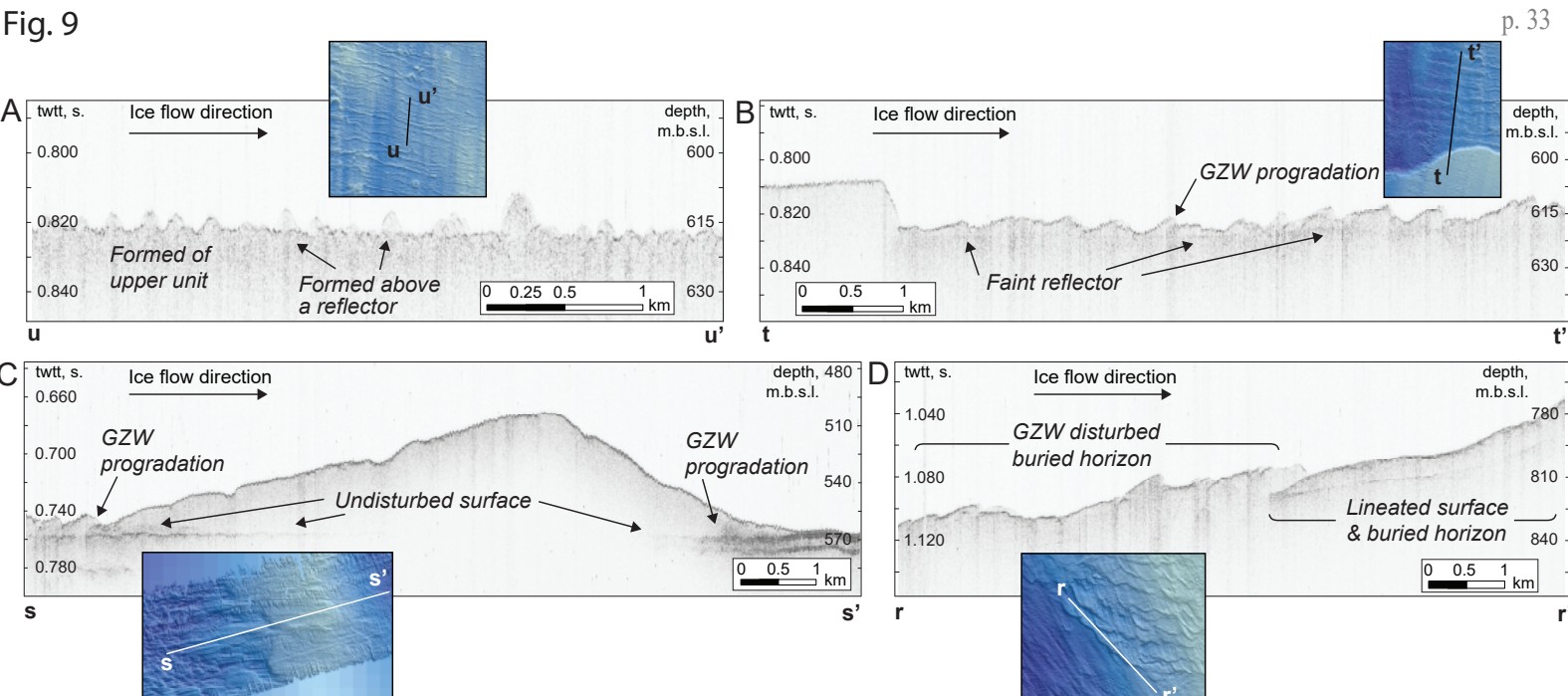





# Fig. 10

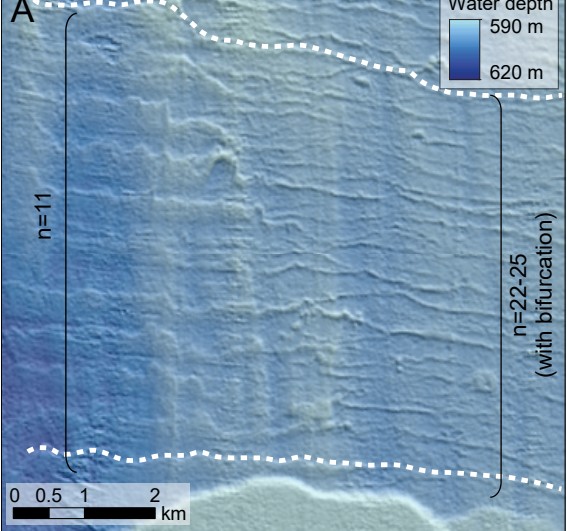

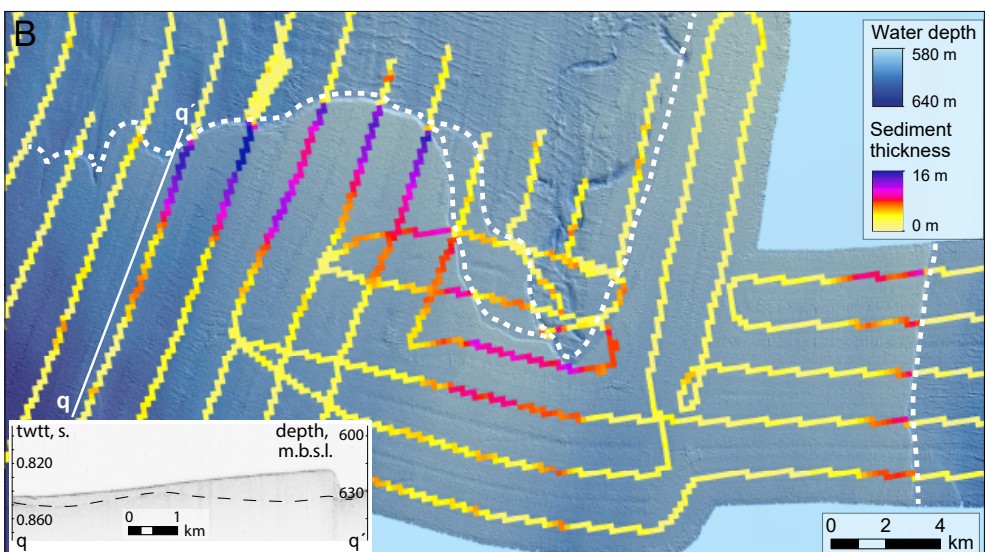





# Fig. 11

5 m

Acoustic profile across GZW

Birds-eye view of GZW surface

Slumps on foreset

50 m



## Fig. 12

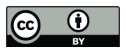



Fig. 13

**A**

**Irregular, inconsistent grounding line retreat buffered from regular forcing**
Back-stepping grounding zone wedges

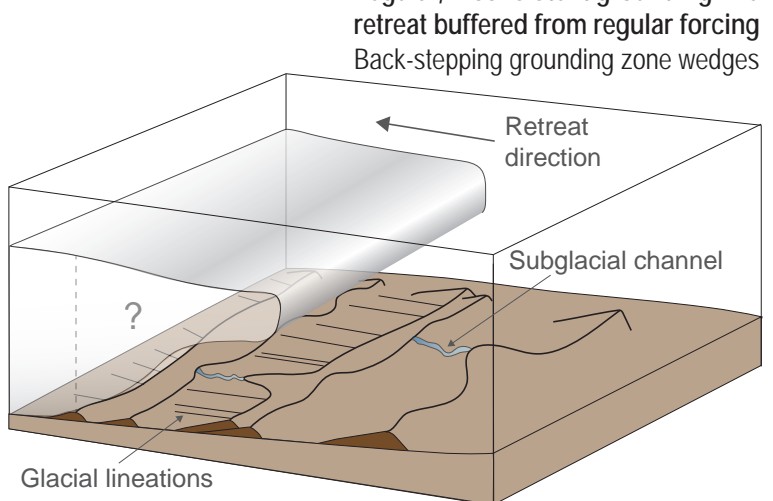
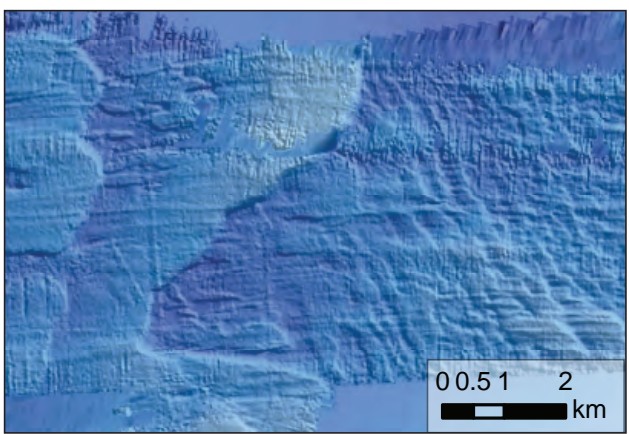

**B**

**Regularly-forced, consistent grounding line retreat**
Recessional moraine field

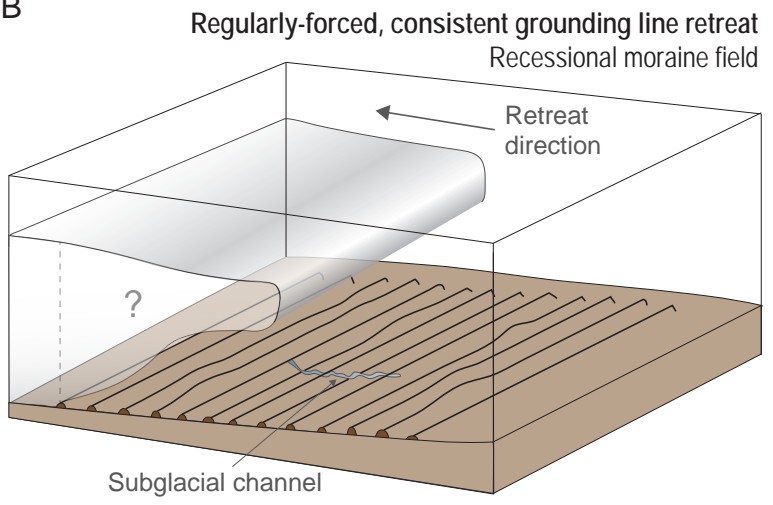
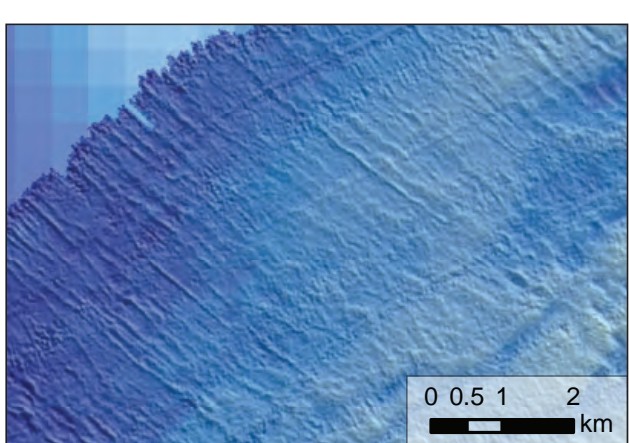

**C**

Landform feedbacks on grounding line stability and instability
Grounding zone wedge asymmetry (stabilising trait) and sinuosity (destabilising trait)

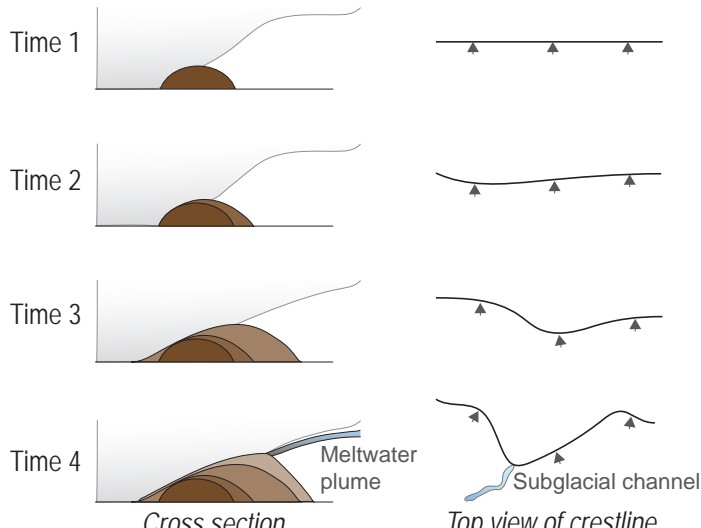
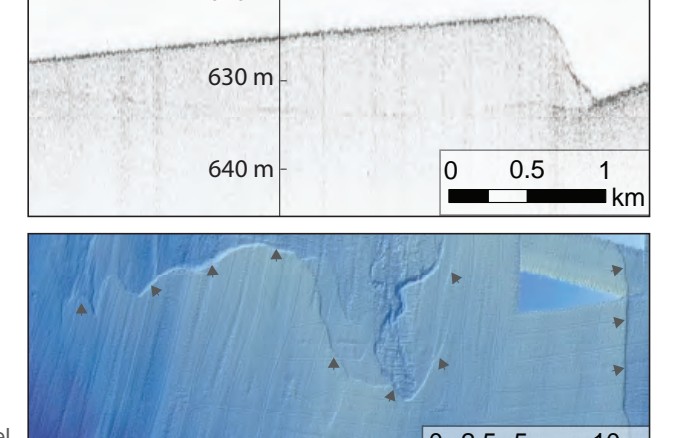

*Cross section*    *Top view of crestline*