# Peer review of "Diagnosing ice sheet grounding line stability from landform morphology"

_The Cryosphere, 2018_

## Referee Comment (RC1) · Anonymous Referee #1 · 21 May 2018

Overview

This paper presents a detailed analysis of the morphology of a large number of grounding line landforms from the western Ross Sea. It is well-illustrated and offers original insights into grounding line processes and controls on GZW and moraine formation. These results will be of broad interest to researchers in the fields of glacial geomorphology and palaeo-glaciology. The main limitation of the paper in its current form is the length of sections 4 and 5, which should be reduced in order to emphasise the key findings. I recommend that the manuscript undergoes revision prior to publication.

Main comments

1) The length of sections 4 and 5 detracts from the key findings of the work. There is

some repetition both within and between sub-chapters, and some interpretations are better-supported than others. Some examples are highlighted below.

2) The paper is generally well-written but there are some confusing sentences and repetition. In particular, the abstract includes several grammatical errors and unclear sentences.

3) Some discussion of flow velocity (ice stream vs. inter-ice stream flow) would be useful in the context of sediment flux. GZWs have been noted to have a strong association with cross-shelf troughs/ former regions of fast ice flow (e.g. Batchelor and Dowdeswell, 2015). Could GZWs be produced preferentially by faster-flowing ice? This could relate to the point about sediment flux, and could help to explain the existence of large terminal moraines that are produced by low sediment flux and long still-stand duration. It is interesting to note that the landforms in the study area tend to group into clusters (corridors?) of related landforms. Is it possible that these mark the former locations of fast and slow-flowing regions of ice, perhaps transient corridors that developed during regional deglaciation?

4) There should be further discussion of recessional moraines in other locations. Symmetry doesn't appear to be a defining characteristic of all recessional moraines, with some reported to display asymmetry with steeper ice-proximal sides. E.g. some of the larger moraines in Todd et al., 2007; Fig. 2 of Lindén and Möller, 2005 shows an asymmetric De Geer moraine. Flink et al. (2016 in Atlas of submarine glacial landforms) suggest that the asymmetry of recessional moraines in Svalbard may indicate their formation by ice-marginal push. It is also interesting to note that recessional moraines of similar dimensions and geometry have been recorded from the terrestrial environment, whereas GZWs appear to only be produced at the margins of marine-terminating ice. Does this lend support to the ice shelf/ ice cliff theory and/or relate to your ideas about grounding line stability?

Additional comments
Abstract

- The first sentence of the abstract is confusing. Surely the grounding line is the point where the ice sheet meets the ocean, not the ice sheet flux? Also remove the comma after 'environments.'

- Line 11. Change to 'the grounding line.'

- Line 13. Change to 'The population is divided into two distinct morphotypes by their morphological properties', or similar.

- Line 19. 'time for which a grounding line is occupied.' This is rather convoluted, perhaps rephrase to 'duration of grounding line occupation'.

- Lines 20 – 23. This sentence is a bit confusing. Isn't the main argument that moraines are associated with 'stable' retreat and GZWs are associated with 'unstable' retreat? Please clarify.

- Lines 24 and 25. 'Short-lived grounding line positions manifest as recessional moraine back-step with small magnitude retreat events'. Please clarity and rephrase.

Introduction

- Page 1, Line 29. The word 'grounded' isn't needed in this sentence.

- Page 2, Line 33; Page 3, Line 2. You mention 'terminal' moraines here and 'recessional' moraines later, without explaining why you switch terminology.

- Page 3, Line 6. 'low profile' of GZWs. Be clearer about this. They are referred to as 'higher amplitude' in the abstract and elsewhere. Would considering the length: height ratios of the landforms help to describe the more wedge-like appearance of GZWs?

- Page 3, Line 22. Change 'whose production is' to 'the production of which is'.

- Page 3, Lines 14 – 33. This section details some theories of the controls on GZW vs. moraine formation. You should also mention the global distribution of GZWs, which

appears to be strongly associated with the sites of formerly fast-flowing sections of ice (i.e. cross-shelf troughs and fjords, e.g. Batchelor and Dowdeswell, 2015). GZW are also only formed in the marine environment.

3. Grounding line landform morphology

- Page 4, Line 28. Change to 'are occasionally.'

- Page 4, Line 31. Add some references for crevasse squeeze ridges (e.g. Ottesen and Dowdeswell, 2006 and references within). - Page 5, Line 13. Change to 'grounding zone wedges in general are found to be more variable in size, sinuosity and asymmetry compared to the. . .' to avoid repetition.

4.1 Topographic setting

This section is long and contains some repetition, which serves to hide the interesting main points that are being made.

For example, Page 6, Line 22: 'suggesting water depth alone does not dictate the formation of a particular landform' and Page 6, Line 26: 'again implies that water depth has little direction influence on the type of grounding line landform.' This point is made yet again in Page 6, Lines 26-27: 'we question, therefore, whether water depth has an influence on landform-building processes.'

Another example is Page 7, Lines 2-3: 'grounding zone wedges more commonly follow slope contours' and Page 7, Lines 6-7: 'grounding zone wedges more commonly adjust orientation to slope contours.'

This level of repetition is not necessary considering that these points are summed up concisely in Section 4.4.

- Page 6, Line 19. Add 'which is' before 'not conducive to.'

4.2.1. Sedimentation mechanism

[Figure]

- This section should refer to the fact that some other recessional moraines have been reported to have asymmetry. Perhaps this has something to do with the amount of forward motion of the ice/ ice push?

- Page 8, Lines 19-21. Consider adding a caveat to this statement. Could the lack of these meltwater-related features relate to the climatic regime, which is colder in Antarctica compared with other locations in which these features have been reported? Could this also be an issue of resolution?

4.2.2 Sediment flux and duration

- Some discussion of ice velocity (ice stream vs. inter ice stream locations) should be included in this section. Could a difference in ice velocity explain why the sediment flux at the grounding line position is higher for grounding zone wedges than for recessional moraines?

- This section is an example of where an interesting point, e.g. that there is a difference in sediment flux between the landforms, is made multiple times within a sub-chapter. E.g. Lines 24-26, Line 27, Line 23, Lines 29-30.

- Page 9, Line 6. Remove comma.

- Page 9, Lines 10-11. This sentence is unclear. Perhaps rephrase to 'GZWs are characterised by. . .'

- Page 9, Lines 16-17. Change to 'A paired group of grounding zone wedges and recessional moraines, where grounding zone wedges transition to recessional moraines (Fig. 6), allows us to isolate the time factor of sediment accumulation.'

- Page 10, Lines 9-11. Are proximal fans more likely to develop in more meltwater-dominated environments?

- Page 10, Line 25. Is asymmetric atypical of moraines beyond those in the study area?

- Page 10, Lines 25 – 28. This is an important point which should be addressed

further. Include an example of a large moraine in the marine environment, e.g. the Skjoldryggen moraine ridge on the mid-Norwegian shelf (Rise et al., 2005; Ottesen et al., 2005). It has been suggested that large moraines are typically found in inter-ice stream locations that are characterised by relatively low full-glacial sedimentation rates.

4.3 Presence or absence of an ice shelf

- From Fig. 5, it seems as though those GZWs that reach higher amplitudes than moraines are particularly wide in the ice-flow direction. Vertical accommodation space below an ice shelf increases away from the grounding line. As a caveat, could a GZW therefore 'grow' higher at its ice-distal point compared with its most ice-proximal point?

- Page 11, Line 9. Does this contradict Lines 25-26? Please clarify.

- Page 11, Lines 15 – 22. Consider removing this section as it is inconclusive and doesn't add to the argument.

4.4 Discussion of controls on landform morphology

- Consider shortening the paragraph from Page 11, Line 24 to Page 12, Line 11, which essentially summarises the points made in the preceding sub-chapters.

5. Implications for grounding line (in)stability

- This chapter should be shortened in order to emphasise the most interesting and conclusive arguments. E.g. Page 13, Lines 28-29 isn't needed as this is already stated in Lines 25-26.

- Page 12, Lines 22 – 29. Shorten or remove this section, focusing on the definition of stability that is used in this paper.

- Page 13, Lines 11 - 13. This is an interesting point. Could it relate to ice velocity? I.e. do ice streams tend to have a more 'unstable'/ episodic style of retreat compared with slower-flowing areas?

Figures

- The landforms in several of the figures need to be labelled or arrowed. E.g. the moraines in Fig. 1B; moraines in Fig. 3A and B, crevasse squeeze ridges in 3H and I; moraines/ GZWs in Fig. 12C and D; moraines/ GZWs in Fig. 13A-C.

- Figure 2 needs to more clearly show the depth of the seafloor, either by using a different colour scheme or by showing some depth contours. The seafloor depth and locations of the troughs/ banks are not clear at present.

---

## Referee Comment (RC2) · Anonymous Referee #2 · 14 Jun 2018

General Comment

This is a well written, well illustrated and very interesting paper that investigates the morphology of grounding-line landforms in the western Ross Sea, Antarctica, and discusses their implications for grounding line retreat and controls thereon. The paper is very suitable for The Cryosphere and will be of particular interest to glacial geomorphologists and paleo-glaciologists but should also be of interest to glaciologists working on grounding-line dynamics and controls. Overall the paper is strong but there are a few points that the authors should address prior to publication (see below).

Specific Comments

1. There needs to be a greater discussion of these grounding line landforms as found in

other glacimarine environments, particularly associated with tidewater glaciers in temperate glacimarine environments such as SE Alaska. This is important as the present paper argues that the specific type of grounding line landform (moraine or grounding zone wedge (GZW)) is independent of the type of glacier front (ice shelf vs grounded tidewater margin). Moraines similar to those described in the present paper have been documented in temperate glacimarine environments but have GZWs? If not then it might suggest that GZWs are preferentially associated with ice shelves?

2. P. 4 lines 26-29. You mention that GZWs are occasionally overprinted by glacial lineations but the latter are never associated with the moraines. Can you clarify exactly what you mean by "associated with"? Do you mean incised over the tops of the moraines or terminating against the proximal face of the ridge or...? It is interesting to consider the morphology of the moraine ridges if they were to be overridden. Presumably they would be smeared out and overprinted by lineations (to some degree at least). Would you be able to differentiate these overridden moraines from GZWs?

3. P. 4 lines 29-30. You infer the presence of crevasse squeeze ridges but say relatively little about them. Such features are commonly associated with surging glaciers in both terrestrial and marine settings and indeed are often regarded as a particularly diagnostic element of the surging glacier landsystem (e.g., Evans and Rea, 1999 Annals of Glaciology; Ottesen and Dowsdeswell, 2006, JGR). Are such features usually found in association with paleo-ice streams elsewhere and could their presence indicate some form of change to flow dynamics?

4. On p. 7 lines 30-32 you go on to say that the crevasse squeeze ridges are evidence for the "squeeze of subglacial sediment upward into the vacant space at the ice base...". I think the latter could be reworded a little clearer – e.g., "...into basal crevasses...".

5. The sentence on p. 6 "We question therefore whether water depth has an influence on landform-building processes" is rather sweeping. Surely it will do where the ice

sheet retreats rapidly on a reverse bed slope and so precluding the formation of such landforms in the first place?

6. I think section 5 'Implications for grounding line stability' could be reduced in length without detriment to the paper. For example I think the introductory paragraph on p. 12 could either be cut or shortened.

---

## Editor Comment (EC1) · C. R. Stokes (Editor) · 25 Jun 2018

I would like to put on record my thanks to the two reviewers for their careful and constructive comments. It is clear that they are very positive, but both suggest some areas that might be further improved. I would therefore like to encourage the authors to consider submitting a revised version of their manuscript.

---

## Author Comment (AC1) · 12 Jul 2018

We thank the reviewers and editor for constructive comments, which we respond to collectively in the supplemental file 'tc-2018-44_AC_12Jul.pdf'. Also included in the supplement are clean and track-changes copies of the revised manuscript and updated figures.

Please also note the supplement to this comment:
https://www.the-cryosphere-discuss.net/tc-2018-44/tc-2018-44-AC1-supplement.zip

---

## Author Response (AR1)

Overview
This paper presents a detailed analysis of the morphology of a large number of grounding line landforms from the western Ross Sea. It is well-illustrated and offers original insights into grounding line processes and controls on GZW and moraine formation. These results will be of broad interest to researchers in the fields of glacial geomorphology and palaeo-glaciology. The main limitation of the paper in its current form is the length of sections 4 and 5, which should be reduced in order to emphasise the key findings. I recommend that the manuscript undergoes revision prior to publication.

Main comments
1) The length of sections 4 and 5 detracts from the key findings of the work. There is some repetition both within and between sub-chapters, and some interpretations are better-supported than others. Some examples are highlighted below. *We have edited Sections 4 & 5 in particular to shorten text and to focus on and clarify key findings.*
2) The paper is generally well-written but there are some confusing sentences and repetition. In particular, the abstract includes several grammatical errors and unclear sentences. *We have revised the abstract, and have edited the text throughout with a view to shortening and removing unnecessary repetition.*
3) Some discussion of flow velocity (ice stream vs. inter-ice stream flow) would be useful in the context of sediment flux. GZWs have been noted to have a strong association with cross-shelf troughs/ former regions of fast ice flow (e.g. Batchelor and Dowdeswell, 2015). Could GZWs be produced preferentially by faster-flowing ice? This could relate to the point about sediment flux, and could help to explain the existence of large terminal moraines that are produced by low sediment flux and long still-stand duration. It is interesting to note that the landforms in the study area tend to group into clusters (corridors?) of related landforms. Is it possible that these mark the former locations of fast and slow-flowing regions of ice, perhaps transient corridors that developed during regional deglaciation?
*Velocity is a relevant issue both in the context of sediment fluxes and in the context of topographic relationships (troughs = streams, for example). One of our main arguments is that grounding zone wedges and moraines cannot solely be differentiated by streaming/non-streaming ice based on lateral transitions between morphotypes within the same trough. Temporally transient corridors of fast/slow flowing ice could potentially result in switches from moraine to grounding zone wedge clusters within a retreat sequence yet one might expect to see other landform evidence of dramatically varying flow. We have modified the text to improve clarity of our findings and the issue of streaming/non-streaming flow as a potential control on landform morphotype.*
4) There should be further discussion of recessional moraines in other locations. Symmetry doesn't appear to be a defining characteristic of all recessional moraines, with some reported to display asymmetry with steeper ice-proximal sides. E.g. some of the larger moraines in Todd et al., 2007; Fig. 2 of Lindén and Möller, 2005 shows an asymmetric De Geer moraine. Flink et al. (2016 in Atlas of submarine glacial landforms) suggest that the asymmetry of recessional moraines in Svalbard may indicate their formation by ice-marginal push. It is also interesting to note that recessional moraines of similar dimensions and geometry have been recorded from the terrestrial environment, whereas GZWs appear to only be produced at the margins of marine-terminating ice. Does this lend support to the ice shelf/ ice cliff theory and/or relate to your ideas about grounding line stability?

*We have added discussion of asymmetric moraines and the formation by push at the grounding line, the occurrence of similar terrestrial moraines, and an expanded discussion of these topics in relation to ice shelf/cliff presence.*

Additional comments
*Abstract:*
*We have revised the abstract.*
- The first sentence of the abstract is confusing. Surely the grounding line is the point where the ice sheet meets the ocean, not the ice sheet flux? Also remove the comma after 'environments.'
- Line 11. Change to 'the grounding line.'
- Line 13. Change to 'The population is divided into two distinct morphotypes by their morphological properties', or similar.
- Line 19. 'time for which a grounding line is occupied.' This is rather convoluted, perhaps rephrase to 'duration of grounding line occupation'.
- Lines 20 – 23. This sentence is a bit confusing. Isn't the main argument that moraines are associated with 'stable' retreat and GZWs are associated with 'unstable' retreat? Please clarify.
- Lines 24 and 25. 'Short-lived grounding line positions manifest as recessional moraine back-step with small magnitude retreat events'. Please clarity and rephrase.
*With respect to these final two points, the main argument is not that one landform type can be considered indicative of 'stable' and the other 'unstable' retreat, but that 'stability as duration' and 'stability as retreat magnitude' are not the same thing. We have clarified this argument both in rewriting the abstract and in the later text.*

*Introduction*
- Page 1, Line 29. The word 'grounded' isn't needed in this sentence. *Removed*
- Page 2, Line 33; Page 3, Line 2. You mention 'terminal' moraines here and 'recessional' moraines later, without explaining why you switch terminology. *We remove the distinction in this sentence (merely 'moraines') and clarify terminology in Section 1 paragraph 4 and in the first paragraph of Section 3.*
- Page 3, Line 6. 'low profile' of GZWs. Be clearer about this. They are referred to as 'higher amplitude' in the abstract and elsewhere. Would considering the length: height ratios of the landforms help to describe the more wedge-like appearance of GZWs? *After consideration, 'low profile' is not necessary and has been removed.*
- Page 3, Line 22. Change 'whose production is' to 'the production of which is'. *Done*
- Page 3, Lines 14 – 33. This section details some theories of the controls on GZW vs. moraine formation. You should also mention the global distribution of GZWs, which appears to be strongly associated with the sites of formerly fast-flowing sections of ice (i.e. cross-shelf troughs and fjords, e.g. Batchelor and Dowdeswell, 2015). GZW are also only formed in the marine environment. *Done*

*3. Grounding line landform morphology*
- Page 4, Line 28. Change to 'are occasionally.' *Done*
- Page 4, Line 31. Add some references for crevasse squeeze ridges (e.g. Ottesen and Dowdeswell, 2006 and references within). *Done*
- Page 5, Line 13. Change to 'grounding zone wedges in general are found to be more variable in size, sinuosity and asymmetry compared to the: : :' to avoid repetition. *Done*

*4.1 Topographic setting*
This section is long and contains some repetition, which serves to hide the interesting main points that are being made. For example, Page 6, Line 22: 'suggesting water depth alone does not dictate the formation of a particular landform' and Page 6, Line 26: 'again implies that water depth has little direction influence on the type of grounding line landform.' This point is made yet again in Page 6, Lines 26-27: 'we question, therefore, whether water depth has an influence on landform-building processes.'

Another example is Page 7, Lines 2-3: 'grounding zone wedges more commonly follow slope contours' and Page 7, Lines 6-7: 'grounding zone wedges more commonly adjust orientation to slope contours.'
This level of repetition is not necessary considering that these points are summed up concisely in Section 4.4.
*Text has been shortened and unnecessary repetition removed.*
- Page 6, Line 19. Add 'which is' before 'not conducive to.' *Edited sentence (with 'that is').*

*4.2.1. Sedimentation mechanism*
- This section should refer to the fact that some other recessional moraines have been reported to have asymmetry. Perhaps this has something to do with the amount of forward motion of the ice/ice push? *Done.*
- Page 8, Lines 19-21. Consider adding a caveat to this statement. Could the lack of these meltwater-related features relate to the climatic regime, which is colder in Antarctica compared with other locations in which these features have been reported? Could this also be an issue of resolution? *This is not a function of data resolution - we can see metre-scale landforms including numerous meltwater channels. We would see fan lobes if they existed; in fact they do exist in other (comparable resolution) data collected from Antarctica. The ubiquity of meltwater channels in the western Ross Sea (Simkins et al. 2017b, Lee et al. 2017, Greenwood et al. accepted) suggests the lack of glaciofluvial fans is not due to lack of subglacial meltwater (Antarctica's climate regime) but could certainly be due to a different hydrological mode/regime that conveys that water (& entrained sediments) to the grounding line. The development of embayments instead of fans suggests we have a lack of meltwater* deposition*, rather than a lack of meltwater.*

*4.2.2 Sediment flux and duration*
- Some discussion of ice velocity (ice stream vs. inter ice stream locations) should be included in this section. Could a difference in ice velocity explain why the sediment flux at the grounding line position is higher for grounding zone wedges than for recessional moraines? *Done.*
- This section is an example of where an interesting point, e.g. that there is a difference in sediment flux between the landforms, is made multiple times within a sub-chapter. E.g. Lines 24-26, Line 27, Line 23, Lines 29-30. *Text has been edited to remove unnecessary repetition.*
- Page 9, Line 6. Remove comma. *Done*
- Page 9, Lines 10-11. This sentence is unclear. Perhaps rephrase to 'GZWs are characterised by: : :' *Rephrased.*
- Page 9, Lines 16-17. Change to 'A paired group of grounding zone wedges and recessional moraines, where grounding zone wedges transition to recessional moraines (Fig. 6), allows us to isolate the time factor of sediment accumulation.' *Done*
- Page 10, Lines 9-11. Are proximal fans more likely to develop in more meltwater dominated environments? *The western Ross Sea is a meltwater-rich environment, as evidenced by ubiquitous palaeo-subglacial channels incised, yet proximal fans have not been observed. Rather, embayments in grounding line landforms are present where channels have delivered water to grounding lines (Simkins et al., 2017b). This indicates that features (e.g. fans, embayments) at the terminus of channels are strongly controlled by other factors, such as basal thermal conditions and sediment rheology.*
- Page 10, Line 25. Is asymmetric atypical of moraines beyond those in the study area? *Addressed.*
- Page 10, Lines 25 – 28. This is an important point which should be addressed further. Include an example of a large moraine in the marine environment, e.g. the Skjoldryggen moraine ridge on the mid-Norwegian shelf (Rise et al., 2005; Ottesen et al., 2005). It has been suggested that large moraines are typically found in inter-ice stream locations that are characterised by relatively low full-glacial sedimentation rates. *Done.*

*4.3 Presence or absence of an ice shelf*

- From Fig. 5, it seems as though those GZWs that reach higher amplitudes than moraines are particularly wide in the ice-flow direction. Vertical accommodation space below an ice shelf increases away from the grounding line. As a caveat, could a GZW therefore 'grow' higher at its ice-distal point compared with its most ice-proximal point? *Yes, it is true that an ice shelf cavity will increase in depth farther from the grounding line; however, the distance from any subglacial sediment source also increases. Grounding zone wedge construction occurs at/up to the proximal point (noted by the position of the grounding line) and then prograde to a seemingly distal point, but sedimentation does not occur simultaneously at the proximal and distal points spanning the full along-flow width of the landform. If this were the case, we would have to invoke a sediment mechanism capable of transporting more material into the distal ice point (in the water column) than the proximal ice point where the grounding line is actually located, whereas sediment should be concentrated closest to the source at the grounding line proper.*

- Page 11, Line 9. Does this contradict Lines 25-26? Please clarify. *No, it doesn't contradict: we distinguish between a control on grounding, and a control on the type of landform product that results from grounding.*
- Page 11, Lines 15 – 22. Consider removing this section as it is inconclusive and doesn't add to the argument. *Cut.*

*4.4 Discussion of controls on landform morphology*
- Consider shortening the paragraph from Page 11, Line 24 to Page 12, Line 11, which essentially summarises the points made in the preceding sub-chapters. *Text has been shortened, but some summary is maintained in order to synthesise key aspects of our data.*

*5. Implications for grounding line (in)stability*
- This chapter should be shortened in order to emphasise the most interesting and conclusive arguments. E.g. Page 13, Lines 28-29 isn't needed as this is already stated in Lines 25-26. *Text has been shortened and repetition removed.*
- Page 12, Lines 22 – 29. Shorten or remove this section, focusing on the definition of stability that is used in this paper. *We choose to keep a shortened version of the paragraph mentioned by the reviewer, since these different facets of the concept of 'stability' are an important element of one of our key conclusions - that the duration of grounding line position occupation signifies 'stability' in the opposite sense to the magnitude & regularity of retreat events.*
- Page 13, Lines 11 - 13. This is an interesting point. Could it relate to ice velocity? I.e. do ice streams tend to have a more 'unstable'/ episodic style of retreat compared with slower-flowing areas? *If paleo-ice streams are identified by cross-shelf troughs, then no we cannot relate exclusive gzw or moraine presence to ice streams/flow velocity, as both moraines and gzws are located within paleo-troughs and even adjacent to each other along the same paleo-grounding line.*

*Figures*
- The landforms in several of the figures need to be labelled or arrowed. E.g. the moraines in Fig. 1B; moraines in Fig. 3A and B, crevasse squeeze ridges in 3H and I; moraines/ GZWs in Fig. 12C and D; moraines/ GZWs in Fig. 13A-C.
*Labels have been added or captions have been revised to improve clarity of the figures, except in Fig. 13 as the figures show snapshots of multibeam shown in previous figures and the caption indicates the landform morphotypes in each panel.*
- Figure 2 needs to more clearly show the depth of the seafloor, either by using a different colour scheme or by showing some depth contours. The seafloor depth and locations of the troughs/ banks are not clear at present.
*The color depth range represents the full range of water depths in the western Ross Sea, so that it can be compared directly with Fig. 7A-C. Contours would help visualize changes in water depth, but results in obscuring the landform mapping. Therefore, we have decided to leave Fig. 2 unchanged to preserve the visibility of the landform mapping.*

**Anonymous Referee #2**

General Comment
This is a well written, well illustrated and very interesting paper that investigates the morphology of grounding-line landforms in the western Ross Sea, Antarctica, and discusses their implications for grounding line retreat and controls thereon. The paper is very suitable for The Cryosphere and will be of particular interest to glacial geomorphologists and paleo-glaciologists but should also be of interest to glaciologists working on grounding-line dynamics and controls. Overall the paper is strong but there are a few points that the authors should address prior to publication (see below).

Specific Comments
1. There needs to be a greater discussion of these grounding line landforms as found in other glacimarine environments, particularly associated with tidewater glaciers in temperate glacimarine environments such as SE Alaska. This is important as the present paper argues that the specific type of grounding line landform (moraine or grounding zone wedge (GZW)) is independent of the type of glacier front (ice shelf vs grounded tidewater margin). Moraines similar to those described in the present paper have been documented in temperate glacimarine environments but have GZWs? If not then it might suggest that GZWs are preferentially associated with ice shelves? *As we are focusing on ice sheet margins on continental shelves, we have not added discussion of outlet glaciers in fjord/tidewater settings. We base our ice shelf/cliff interpretations on observations from the western Ross Sea dataset, where landform morphotype is not conclusively linked to the presence/absence of an ice shelf.*

2. P. 4 lines 26-29. You mention that GZWs are occasionally overprinted by glacial lineations but the latter are never associated with the moraines. Can you clarify exactly what you mean by "associated with"? Do you mean incised over the tops of the moraines or terminating against the proximal face of the ridge or: : :? It is interesting to consider the morphology of the moraine ridges if they were to be overridden. Presumably they would be smeared out and overprinted by lineations (to some degree at least). Would you be able to differentiate these overridden moraines from GZWs? *By 'associated with' we mean overprinted or terminating against the proximal side of the ridge - this has been clarified in the text. The morphology of moraines that have been overprinted would potentially be asymmetric, but not necessarily overprinted by lineations considering we see numerous clearly prograding grounding zone wedges without lineated topsets. As we suggest that moraines are the proto-feature to grounding zone wedges, an overridden moraine could in fact be the transitional landform between the two morphotypes, yet manifest visually as a grounding zone wedge rather than a moraine.*

3. P. 4 lines 29-30. You infer the presence of crevasse squeeze ridges but say relatively little about them. Such features are commonly associated with surging glaciers in both terrestrial and marine settings and indeed are often regarded as a particularly diagnostic element of the surging glacier landsystem (e.g., Evans and Rea, 1999 Annals of Glaciology; Ottesen and Dowsdeswell, 2006, JGR). Are such features usually found in association with paleo-ice streams elsewhere and could their presence indicate some form of change to flow dynamics? *As we have only one small patch of crevasse squeeze ridges compared to thousands of grounding line landforms, we do not place much focus on the crevasse squeeze ridges. Crevasse squeeze ridges have been observed in other ice stream setting such as in Bjørnøyrenna Trough in the Barents Sea (Ruther et al., 2013) and Abbot Trough in the Amundsen Sea (Klages et al., 2015). We mention that these ridges could indicate conditions suitable for deformation near the grounding line in Section 4.2.1, and it is possible their presence could indicate a change in flow dynamics and/or change in basal conditions (e.g. till rheology, thermal conditions); yet, do not seem to change the pattern of retreat or landform expression at the grounding line.*

4. On p. 7 lines 30-32 you go on to say that the crevasse squeeze ridges are evidence for the "squeeze of subglacial sediment upward into the vacant space at the ice base: : :". I think the latter could be reworded a little clearer – e.g., ": : :into basal crevasses: : :". *Done*

5. The sentence on p. 6 "We question therefore whether water depth has an influence on landform-building processes" is rather sweeping. Surely it will do where the ice sheet retreats rapidly on a reverse bed slope and so precluding the formation of such landforms in the first place? *This muddles the ability to ground with the type of product that results. Water depth must fundamentally affect the ability to ground. It does not appear, from our data, to govern the type of landform that is built. This sentence has in any case been removed during editing/shortening.*

6. I think section 5 'Implications for grounding line stability' could be reduced in length without detriment to the paper. For example I think the introductory paragraph on p. 12 could either be cut or shortened. *We have edited and shortened the text throughout Sections 4 & 5. We choose to keep a shortened version of the paragraph mentioned by the reviewer, since these different facets of the concept of 'stability' are an important element of one of our key conclusions - that the duration of grounding line position occupation signifies 'stability' in the opposite sense to the magnitude & regularity of retreat events.*

[revised manuscript text omitted]